# The Diagnostic Value of Circulating Cell-Free HPV DNA in Plasma from Cervical Cancer Patients

**DOI:** 10.3390/cells11142170

**Published:** 2022-07-11

**Authors:** Sara Bønløkke, Magnus Stougaard, Boe Sandahl Sorensen, Berit Bargum Booth, Estrid Høgdall, Gitte-Bettina Nyvang, Jacob Christian Lindegaard, Jan Blaakær, Jesper Bertelsen, Katrine Fuglsang, Mikael Lenz Strube, Suzan Lenz, Torben Steiniche

**Affiliations:** 1Department of Clinical Medicine, Aarhus University, 8200 Aarhus, Denmark; magnstou@rm.dk (M.S.); boesoere@rm.dk (B.S.S.); steiniche@clin.au.dk (T.S.); 2Department of Pathology, Aarhus University Hospital, 8200 Aarhus, Denmark; jesber@rm.dk; 3Department of Clinical Biochemistry, Aarhus University Hospital, 8200 Aarhus, Denmark; 4Department of Obstetrics and Gynecology, Aarhus University Hospital, 8200 Aarhus, Denmark; berit.booth@auh.rm.dk (B.B.B.); katrfugl@rm.dk (K.F.); 5Department of Pathology, Herlev Hospital, 2730 Herlev, Denmark; estrid.hoegdall@regionh.dk; 6Department of Oncology, Odense University Hospital, 5000 Odense, Denmark; gitte-bettina.nyvang@rsyd.dk; 7Department of Oncology, Aarhus University Hospital, 8200 Aarhus, Denmark; jacob.lindegaard@auh.rm.dk; 8Department of Obstetrics and Gynecology, Odense University Hospital, 5000 Odense, Denmark; jan.blaakaer@rsyd.dk; 9Department of Clinical Research, University of Southern Denmark, 5230 Odense, Denmark; 10DTU Bioengineering, Technical University of Denmark, 2800 Kongens Lyngby, Denmark; milst@dtu.dk; 11Private Gynecological Clinic “Suzan Lenz Gynækolog”, 2400 Copenhagen, Denmark; suzanlenz@dadlnet.dk

**Keywords:** cervical cancer, HPV, human papillomavirus, digital droplet PCR, ddPCR, circulating cell-free DNA, circulating tumor DNA, ctDNA, circulating cell-free HPV DNA, ccfHPV DNA

## Abstract

Circulating cell-free HPV DNA (ccfHPV DNA) may serve as a marker for cervical cancer. In this study, we used digital droplet PCR (ddPCR) to detect and quantify ccfHPV DNA in plasma from patients with HPV16- or HPV18-associated cervical cancer. Blood samples from 60 patients diagnosed with cervical cancer (FIGO IA1-IVA) at Aarhus or Odense University Hospital (June 2018 to March 2020) were collected prior to treatment, and patients were subdivided into an early stage (*n* = 30) and a late-stage subgroup (*n* = 30) according to disease stage. Furthermore, blood samples from eight women with HPV16- or 18-associated premalignant conditions (CIN3), and 15 healthy controls were collected. ddPCR was used to analyze plasma from all participants. ccfHPV DNA was detected in 19 late-stage patients (63.33%), 3 early stage patients (10.00%), and none of the CIN3 patients or controls. Quantitative evaluation showed significant correlations between ccfHPV DNA level and stage, tumor score, and tumor size. Thus, our results indicate that ccfHPV DNA may not be a useful marker for early detection of cervical cancer. However, for patients with advanced stage cervical cancer, ccfHPV DNA level represents a promising tool to establish tumor burden, making it useful for establishing treatment response and monitoring the disease.

## 1. Introduction

Human papillomavirus (HPV) is the cause of almost all cases of cervical cancer [1], and despite preventive screening and HPV vaccination, cervical cancer ranks as the fourth most frequently diagnosed cancer and the fourth leading cause of cancer death in women worldwide [2]. The most common symptoms of cervical cancer are postcoital bleeding, intermenstrual bleeding, and vaginal discharge [3,4,5,6,7]. However, these symptoms are common in young females with non-malignant conditions, e.g., genital infections [4,8,9].

Consequently, >55% of patients are diagnosed with advanced disease stages (stage II–IV according to International Federation of Gynecology and Obstetrics (FIGO)) [10], and for both patients with stage III–IV disease and the 28–64% of stage IIB-IVA patients experiencing a disease relapse, overall mortality rate is >50% [10].

Thus, there is a need for a valid supplement to the current diagnostic tools regarding both primary disease and not least relapse. A simple, sensitive, and minimally invasive method of growing interest evaluates circulating cell-free DNA (ccfDNA) as a liquid biopsy marker for detecting cancer and monitoring disease [11,12,13]. ccfDNA is fragmented DNA released into the blood by cell death and active secretion processes. In oncology, detection of ccfDNA derived from tumors is known as circulating tumor DNA (ctDNA) [14], and in recent research, circulating cell-free HPV DNA (ccfHPV DNA) has become of major interest as a potential biomarker for early diagnosis and prognosis of cervical cancer [15,16,17]. The HPV genome is a double-stranded circular DNA virus of 8000 bases in length. It encodes six early (E) and non-structural genes; E1, E2, E4, E5, E6, E7 and two late (L) genes; and L1 and L2 forming the viral capsid structure [18]. E6 and E7 are oncoproteins targeting the p53 tumor suppressor protein and the retinoblastoma protein (pRB), respectively. Consequently, the targeted proteins are degraded, leading to dysregulated cell cycle and uncontrolled cell division [19,20]. The HPV genome can exist in the cell as an episome, or it can integrate into the human genome [21]. Earlier studies have shown the pre-sence of integrated HPV DNA in host cells with increasing frequency from normal cervical epithelium to cervical intraepithelial neoplasia (CIN), as well as invasive cancer [22,23,24]. However, some studies have detected episomal HPV DNA in a substantial portion of cervical cancers, either as an episome or as a combination of both integrated and episomal forms [25,26,27,28,29,30]. Altogether, these molecular characteristics explain why HPV-associated tumors represent an obvious model for the analysis of ccfDNA in oncology. The unique viral sequence, the abundance of copies upon subsequent amplification, and the presence of either integrated HPV genomes and/or multiple episomal copies in cancer cells make ccfHPV DNA a favorable marker for detecting cervical cancer [31].

To ensure a precise and not least clinically relevant qualitative and quantitative mea-sure of ccfHPV DNA in blood samples, an assay with sufficient specificity and sensitivity is essential. So far, there has been limited success in detecting ccfHPV DNA in plasma or serum, which has mainly been suggested to be caused by a lack of sensitivity of the me-thods used [32,33,34]. Recent studies have examined the use of digital droplet PCR (ddPCR) to measure ccfHPV DNA [31,35,36,37]. This method is based on partitioning of the sample in many reaction chambers or droplets [38], and it offers better sensitivity and quantification of poorly abundant nucleic acids, including ccfHPV DNA [17,31,32,39,40,41,42,43,44]. In these studies, detection rate ranges between 31% and 100% [31,35,36,37], underlining that findings on ccfHPV DNA in cervical cancer patients and the exact relationship between both its presence and quantity and other clinical parameters are still unclear.

In this study, we performed retrospective analysis on plasma from cervical cancer patients with different disease stages, patients with cervical intraepithelial lesion grade 3 (CIN3), and healthy controls using a highly sensitive ddPCR. With these analyses, we sought to (I) test the diagnostic performance of ccfHPV16 DNA and ccfHPV18 DNA for cervical cancer detection and (II) assess the possible correlation between ccfHPV DNA level and clinical parameters such as patient age, histology, disease stage, tumor score (T score), and tumor size. T-score is not routinely used for cervical cancer patients. The sco-ring system has recently been developed at Aarhus University Hospital [45], and after verification and an extension of the system, it is now based on integrated evidence from both clinical examination and sectional magnetic resonance imaging (MRI) [46]. In contrast to the current FIGO staging system, the T-scoring system adds relevant information regarding tumor invasion into neighboring anatomic locations [47], and it has been shown to be a powerful prognostic factor for local control and survival [45].

## 2. Materials and Methods

This proof-of-concept study is the initial part of a large setup, where all patients diagnosed with cervical cancer between June 2018 and October 2020 at Aarhus University Hospital or between January 2019 and January 2020 at Odense University Hospital were invited to participate (Figure 1). The inclusion criteria for the overall study cohort were as follows: patients ≥ 18 years of age diagnosed at Aarhus or Odense University Hospital with primary cervical cancer, FIGO stage ≥ IB1, and referral to curatively intended treatment. When the study was initiated, staging of cervical cancer was performed with FIGO 2009 classification [10], but from 1 January 2020, staging of cervical cancer in Denmark switched to the FIGO 2018 classification [48]. Thus, to ensure that our study cohort was staged with the same classification system, patients included in 2018 and 2019 with FIGO 2009 were “re-staged” with FIGO 2018, and some were therefore “down-staged” from stage IB to stage IA. The rationale behind not including patients with <stage IB1 was that the great majority of these patients are cured after their surgical procedure, and thus, in Denmark, these patients are not offered a follow-up program for two years at the hospital. Since we planned to collect blood samples from all patients both before, during, and after treatment for up to two years after their diagnosis, this criterion was chosen as a cutoff. However, for patients that were down-staged from stage IB with FIGO 2009 to stage IA with FIGO 2018, these were treated and followed according to their initial staging (FIGO 2009) with a two-year follow-up program, and therefore, these were not excluded from the study cohort. The exclusion criteria were as follows: patients with relapse of a prior cervical cancer, FIGO stage < IB1 (until December 2019 with FIGO 2009) [10], and from January 2020 with FIGO 2018 [48]), and referral to palliative treatment. Furthermore, patients whose cervical biopsies were tested HPV negative were excluded.

For this proof-of-concept study, only patients with either HPV16 or HPV18 positive cervical cancer were included in the case cohort since these genotypes account for appro-ximately 70% of cervical cancer cases [49]. In addition to this, patients with multiple HPV genotypes were not included in this study (Figure 1). The reason for this was that we only tested for either ccfHPV16 DNA or ccfHPV18 DNA in the blood, and for patients with multiple HPV genotypes, the genotype primary responsible for developing and driving the cancer was unknown.

Our aim was to include the first 30 patients with HPV16/18 positive early stage loca-lized cervical cancer (i.e., early stage subgroup) and the first 30 patients with HPV16/18 positive advanced stage cervical cancer (i.e., late-stage subgroup). During the period of June 2018 to March 2020, this case cohort of 60 patients was achieved; 30 patients in the early stage subgroup (FIGO IA1-IB2), who had undergone radical surgery, and 30 patients in the late-stage subgroup (FIGO IB3-IVA), who had received primary radiotherapy consisting of external beam radiotherapy (EBRT) followed by high-dose-rate brachytherapy. During EBRT, concomitant weekly cisplatin 40 mg/m^2^ was given to all late-stage patients except two patients who had a poor performance status. Furthermore, in the attempt to shrink the tumor, two patients received neoadjuvant chemotherapy prior to radiotherapy. Figure 1 provides a flowchart of the overall inclusion of cervical cancer patients. Appendix A provides clinical and biological data on the cohort, and Appendix A provides the causes for exclusion for patients excluded from the study.

Blood samples from all patients were collected at the time of diagnosis and prior to treatment initiation (i.e., surgery, primary radiotherapy, or neoadjuvant chemotherapy). Additionally, blood samples from 25 patients diagnosed with CIN3 and included at Regional Hospital Randers (CIN3) were collected (Appendix A). The reason for including CIN3 patients was to see whether ccfHPV DNA is already detectable in patients with premalignant lesions. Furthermore, for CIN3 patients, only eight were included in the following analyses since the remaining patients had lesions positive for other high-risk HPV genotypes than HPV16 or HPV18 (Appendix A). A control group consisting of 15 healthy anonymous women (negative controls) were included in connection with blood donation at the Department of Clinical Immunology at Aarhus University Hospital. These women all had a negative cervical smear within the past three years.

All CIN3 patients and cervical cancer patients were thoroughly informed and granted written informed consent prior to blood sampling.

### 2.1. DNA Extraction and HPV Analyses on Cervical Tissue

HPV genotyping was performed on cervical tissue from cancer patients and CIN3 patients. From each patient, a formalin-fixed and paraffin-embedded (FFPE) cervical tissue block was collected from the referring private gynecologist or hospital in connection with the diagnosis of the disease. All blocks were sectioned at the Department of Patho-logy, Aarhus University Hospital. For all tissue blocks, three to four 10 μm thick sections were cut and collected in a sterile tube. Subsequently, a 3 μm thick section was cut for hematoxylin and eosin staining (HE) to ensure the presence of preneoplastic (for CIN3 patients) or neoplastic (for cancer patients) tissue in the sections subjected to HPV analysis. DNA extraction was performed using the QIAsymphony DSP DNA Mini Kit, version 1 (Qiagen, Venlo, The Netherlands). HPV DNA was detected by the INNOLiPA HPV Ge-notyping Extra II (INNO-LiPA) (Fujirebio Europe, Ghent, Belgium) according to the ma-nufacturer’s instructions. The INNOLiPA uses the SPF10 primer set to amplify a 65 base pair (bp) region in the L1 gene, which is then followed by HPV genotyping by reverse hybridization. The assay is capable of detecting 32 HPV genotypes, including the high-risk HPV genotypes (16, 18, 31, 33, 35, 39, 45, 51, 52, 56, 58, 59), most of the potentially high-risk HPV genotypes (25, 53, 66, 68, 70, 73, 82), and the low-risk HPV genotypes (6, 11, 40, 42, 43, 44, 54, 61, 62, 67, 81, 83, 89).

### 2.2. Blood Sample Preparation

For ccfHPV DNA detection, a total of 30 mL of full blood was collected from each participant in Cell-Free DNA BCT^®^ collection tubes (Streck, La Vista, NE, USA). Accor-ding to the manufacturer’s instructions, samples were stored at room temperature (6–37 °C) for up to 14 days before DNA extraction. Extraction of cell-free plasma DNA was performed through centrifugation at 1600× *g* for 10 min, followed by an additional centrifugation at 16,000× *g* for 10 min. Plasma samples were stored at −80 °C, and free-circulating DNA was extracted from 5 mL of plasma using QIAamp Circulating Nucleic Acid Kit (QIAGEN, Hilden, Germany) following the manufacturer’s instructions.

### 2.3. ddPCR Analysis for ccfHPV DNA Detection

The isolated cfDNA was assayed for ccfHPV16 or ccfHPV18 DNA detection using the QX200 AutoDG Droplet Digital PCR system (Bio-Rad Laboratories, Inc., Hercules, CA, USA). Prior to the ddPCR analyses on plasma from cases, CIN3 patients, and negative controls, a control analysis for the presence of human DNA in the isolated cfDNA was performed using an assay targeting the human reference gene beta-2-microglobuline (B2M). These analyses confirmed that all cfDNA samples isolated in the study contained human cfDNA (data not shown). Set-up of the following HPV ddPCR reactions was performed with the AutoDG Droplet Generator (Bio-Rad). Each reaction was run in a total volume of 22 µL. The assays included 1× ddPCR Supermix for Probes (No dUTP) (Bio-Rad), forward primer, reverse primer, probe, and purified ccfDNA. The following cycling conditions were used for PCR amplification: 95 °C for 10 min, 40 cycles of 96 °C for 30 s and 62 °C for 1 min, 98 °C for 10 min, and 4 °C for infinite hold. Furthermore, samples were run in triplicate. To ensure the best possible quality of our analyses, each run included a non-template control, cfDNA from a negative control, and an HPV positive control consisting of purified DNA from cervical biopsies from selected cases with HPV16 and cases with HPV18 associated cervical cancer (positive controls). For all analyses, QuantaSoft analysis software version v.1.7.4.0917 (Bio-Rad) was used. Reactions were performed for each HPV genotype (HPV16 and HPV18), giving us one ddPCR assay for ccfHPV16 detection and one for ccfHPV18 DNA detection.

The primers and probe used for the HPV16 ddPCR assay have previously been designed and successfully used for ddPCR-based ccfHPV16 DNA detection of plasma and serum samples [31], whereas the primers and probe used for the HPV18 ddPCR assay were custom-made (Appendix A). Both primer-probe sets were located within the E7 oncogene of HPV because the E6 and E7 sequences are the most highly amplified in tumor genomes [50]. In view of the highly fragmented nature of ccfDNA, the size of the resulting amplicons was short: 83 bps for HPV16 and 71 bps for HPV18.

To validate that our HPV16 and HPV18 assays amplified these two DNA species, we used purified DNA from the already analyzed cervical biopsies from the case cohort with HPV16/18 positive tumor tissue (positive controls) (Figure 2A and Appendix A). Agarose gel electrophoresis demonstrated a band of the expected size for both HPV16 and HPV18. The bands were cut out of the gel and sanger sequenced to confirm the correct identity of the two PCR products. As has previously been described [51], the limit of detection (LoD) was determined by analyzing ccfDNA from the negative controls. ccfDNA isolated from 15 healthy women was used to determine the limit of detection (LoD) of the ddPCR assays. In this group, the highest measurable quantity of ccfHPV16 DNA and ccfHPV 18 DNA was 1.67 copies per mL plasma (Appendix A). Thus, we established a conservative value for the cutoff by using twice the amount of this (3.34 copies per mL plasma) as cutoff, giving us a cutoff value for ccfHPV16/18 DNA positivity of >3 copies/mL.

### 2.4. Statistics

All digital PCR data are presented as log10-transformed. Categorical data were analyzed with the Kruskal–Wallis test and, when significant, followed up with the Conover-Iman test for post hoc analysis. Numerical data were analyzed using linear regression on log-transformed data. All statistical tests were made in R: The R Project for Statistical Computing. R Foundation for Statistical Computing, Vienna, Austria. Available from URL: https://www.r-project.org/ [accessed on 1 July 2022], version 4.1.1. A *p-*value below 0.05 was considered significant.

Detection rate of ccfHPV DNA was defined as the number of ccfHPV DNA positive plasma samples from cancer patients divided by the total number of cancer patients.

## 3. Results

### 3.1. Patient Characteristics, Histology, and HPV Detection in Cervical Tissue Biopsies

The case cohort consisted of 60 HPV16 or HPV18 positive cervical cancer patients (Figure 1), 30 with localized cancer, i.e., the early stage subgroup, and 30 with advanced cancer, i.e., the late-stage subgroup (Table 1 and Appendix A). The control cohort consisted of 15 healthy women (Appendix A). Furthermore, 25 CIN3 patients were included, whereof eight were positive for HPV16 by genotyping (Appendix A). These were therefore included for the following ddPCR analyses. Histologically, the cervical cancer cases consisted of 41 (68.3%) squamous cell carcinomas (SCC), 17 (28.3%) adenocarcinomas (AC), and two (3.3%) adenosquamous carcinomas (ASC) (Appendix A). Of the 60 cervical cancer patients, 48 (80%) had HPV16 positive tumors and 12 (20%) were HPV18 positive by genotyping (Appendix A). Table 1 shows patient and tumor characteristics for the two subgroups, and furthermore, all relevant clinical and biological data are given in Appendix A.

### 3.2. Detection of ccfHPV16 and ccfHPV18 DNA Using ddPCR

All samples were subjected to ccfHPV DNA analysis. First, cfDNA from CIN3 patients was analyzed, and here, ccfHPV DNA was not detected in any of the eight samples (Figure 2A and Appendix A).

For the two subgroups, results on ccfHPV DNA differed significantly (*p* < 0.000) (Figure 2B), and detection rate was therefore calculated for the two subgroups separately. For the late-stage subgroup (*n* = 30), ccfHPV DNA was detected in plasma samples from 19 patients demonstrating a detection rate of 63.3% (Appendix A). All patients positive for ccfHPV DNA were patients with HPV16-associated cancers, and thus when focusing only on HPV16 ddPCR analyses (*n* = 27), the HPV16 genotype-specific detection rate was 70.4%. Interestingly, for HPV18 (*n* = 3), none of the plasma samples were ccfHPV18 DNA positive. For patients in the early stage subgroup (*n* = 30), only three were positive for ccfHPV DNA, demonstrating a detection rate of 10.0% (Appendix A). Furthermore, two of these patients had only four and five HPV copies/mL plasma, respectively. In this subgroup, the HPV16 genotype-specific detection rate was 14.3%, and as with the late-stage subgroup, none of the plasma samples from patients with HPV18-associated cancer (*n* = 8) were ccfHPV18 DNA positive.

### 3.3. Quantification of ccfHPV DNA

ddPCR can provide an absolute quantification of the molecular target [52]. In the late-stage subgroup (*n* = 30), ccfHPV DNA ranged from 0 to 810 copies/mL (Appendix A). The median ccfHPV DNA level was 17.5 (Figure 2B), and the mean ccfHPV DNA level was 84.6 copies/mL (±172.2). The high SD value is related to the highly variability in ccfHPV DNA content among patients. For the early stage subgroup (*n* = 30), ccfHPV DNA ranged from 0 to 13 copies/mL (Appendix A). The median ccfHPV DNA level was 1.0 copy/mL (Figure 2B), and the mean ccfHPV DNA level was only 1.0 copy/mL (±3.3).

### 3.4. ccfHPV DNA Level Correlated to Clinical and Biological Parameters

We examined whether ccfHPV DNA levels correlated with any clinical or biological parameters by comparing the ccfHPV DNA result with patient’s age, histology, FIGO stage, T score [46], and tumor size (Figure 3). In order to establish a possible correlation between ccfHPV DNA level and FIGO 2018 stage (Figure 3C), the stages were subdivided into four groups consisting of stage IA (IA1 + IA2), IB (IB1 + IB2 + IB3), IIB, and IIIC/IV (IIIC1 + IIIC2 + IVA). Here, we found significant correlations between ccfHPV DNA level and FIGO stage (Figure 3C), with the highest ccfHPV DNA levels detected in the more advanced disease stages. Furthermore, a weak, but significant, positive association between ccfHPV DNA level and T score (R^2^ = 0.1, *p* = 0.01) (Figure 3D) and between ccfHPV DNA level and tumor size (R^2^ = 0.3, *p* < 0.005) (Figure 3E) was detected. No association between ccfHPV DNA level and patient age (*p =* 0.1) (Figure 3A) or histology was observed (Figure 3B).

## 4. Discussion

In this study, we examined whether ccfHPV DNA can be used as a marker for early detection of cervical cancer, as well as whether ccfHPV DNA level is correlated with different clinical and biological parameters, which could potentially be used to guide treatment and improve surveillance after treatment. By means of ddPCR, we demonstrated that ccfHPV DNA is mainly detectable in patients with advanced stage disease, and we furthermore found that ccfHPV DNA level correlated with disease stage, T score, and tumor size.

Cervical cancer differs from other cancers because HPV infection is a critical step in carcinogenesis. Thus, detection of the virus in the blood of these patients as ccfHPV DNA could be considered a liquid biopsy with diagnostic and prognostic potential. Earlier stu-dies on ccfHPV DNA in cervical cancer patients differ markedly in terms of sensitivity and correlations with clinical parameters. Several issues may explain these variable fin-dings. Firstly, there is a great variation in the methods used to detect ccfHPV DNA [15,31,32,34,35,36,37,39,40,41,42,43,53]. Since ddPCR offers a very high sensitivity and enables an absolute target quantification of the molecular target [52], this method is the obvious choice to determine the diagnostic performance of ccfHPV DNA for early detection of cervical cancer. For other studies using ddPCR for ccfHPV DNA analysis [31,35,36,37], sensitivities vary between 31% and 100%, and thus the diagnostic performance of ccfHPV DNA remains unclear. Different reasons for these inconclusive results may explain this. First, the mentioned studies vary considerably in the number of patients included, ranging from 19 to 138 patients, making it difficult to compare the statistics. Furthermore, important patient characteristics such as disease stage vary significantly with some studies only including part of the staging spectrum, which also makes it difficult to compare the studies. Another exceedingly important detail is the inclusion of blood samples from healthy negative controls, which is essential to enable the determination of a reliable cutoff for ccfHPV DNA positivity and to eliminate possible viral contamination. Furthermore, the inclusion of a human control gene to ensure the presence of human DNA in a sample is essential to ensure that ccfHPV-negative samples are in fact negative, and overall, only some of the prior studies include these control components. In this study, we used the ddPCR assay to analyze plasma samples from 60 patients with localized or advanced stage cervical cancer (i.e., early stage and late-stage subgroup), 8 CIN3 patients and 15 healthy controls for ccfHPV16 or ccfHPV18 DNA. A human reference control step was conducted to ensure that all plasma samples contained human DNA. By using plasma samples from the healthy controls, the assay was then analytically validated, and we demonstrated a conservative cutoff of ccfHPV DNA positivity of >3 copies/mL plasma. To ensure the best possible quality of our HPV16 and HPV18 ddPCR analyses, we followed the same approach as used in ddPCR assays for clinical diagnostics. Each ddPCR run included a non-template control, cfDNA from a healthy control, and an HPV-positive control consisting of purified DNA from cervical biopsies from selected cases with HPV16 and cases with HPV18 associated cervical cancer (positive controls). To establish whether ccfHPV DNA is already present in patients with premalignant conditions, we first analyzed plasma samples from eight patients with HPV16/18-associated CIN3 and showed that ccfHPV DNA was not detected in these patients. This suggests that tumor DNA is only released to the circulation of patients with invasive cancer, which is in line with previous findings [32,39,40]. ddPCR analyses on cases revealed ccfHPV DNA positivity in 63% of late-stage patients and in 10% of early stage patients. These results indicate that tumor DNA is only released to the circulation in a minority of patients with localized disease and thus low tumor burden, whereas for patients with more advanced stage cervical cancer, ccfHPV DNA is often present. This tendency is in agreement with two of the prior studies [36,37].

Results from the earlier studies using ddPCR for ccfHPV DNA detection in cervical cancer patients [31,35,36,37] show that the level of ccfHPV DNA in plasma may be associated with different clinical parameters such as disease stage [31,35] and tumor size [31]. In the study by Kang et al. [37], serum from 19 patients with metastatic cervical cancer were analyzed, of which nine patients had received tumor-infiltrating lymphocyte (TIL) immunotherapy. From the last-mentioned patients, sequential serum samples were collected during their treatment. Besides detecting ccfHPV DNA in serum samples from all 19 patients at the time of diagnosis, results from the sequential serum samples showed that in three patients, who experienced objective cancer regression after TIL treatment, a transient ccfHPV DNA peak was detected 2–3 days after TIL infusion. This may be interpreted as the tumor DNA being released from dying cancer cells [12]. Furthermore, results from the two patients, who experienced a complete and long-term response to TIL immunotherapy, showed long-term clearance of ccfHPV DNA. These findings indicate that ccfHPV DNA may be a promising tool for early detection of cervical cancer, but they also suggest that the presence and level of ccfHPV DNA may be correlated to tumor dynamics and therefore could be useful for establishing treatment response and monitoring the disease.

In our study, we also examined the possible correlation between ccfHPV DNA level and clinical and biological parameters. We found a significant correlation between ccfHPV DNA level and both disease stage, T score (*p* = 0.01), and tumor size (*p* < 0.005), with tumor size reaching the highest significance. T score is not routinely used for cervical cancer patients, but on the basis of a recent study showing that T-score was a powerful prognostic factor for local control and survival [45], T scores [46] were established for all patients. Here, a weak but significant correlation between ccfHPV DNA level and T score was found, which could indicate that ccfHPV DNA level may also be of prognostic signi-ficance in cervical cancer patients.

In this study, 11 (37%) late-stage patients had non-detectable ccfHPV DNA; four patients with stage IIB disease, six with stage IIIC1, and one with stage IVA. Furthermore, three of the patients had HPV18-associated cancer, and the remaining were HPV16 associated. Many different parameters may explain these ccfHPV DNA negative results. Firstly, a possible limitation of our ddPCR assays concerns sensitivity. With the purpose of achieving a high sensitivity, the amplicons for both the HPV16 ddPCR and the HPV18 ddPCR were short. Furthermore, our ddPCR analyses on tissue from HPV16/18-associated cervical cancer patients (positive controls) validated our ddPCR assays and confirmed that our primers worked sufficiently for both HPV genotypes (Figure 2). For the assays, we used one primer pair and probe for HPV16 and one primer pair and probe for HPV18. It is likely that in some cases of HPV-associated cervical cancer, point mutations or deletions in some regions of the HPV genome may occur. If such genetic changes occur in the sequence where the primers and probe anneal, it could cause false negatives. Cheung at al. [35] increased sensitivity from 55.8% to 61.6% by adding an additional primer pair and probe, and thus it is likely that an addition of primer sets to cover multiple parts of the HPV genome may increase the detection rate. In the present study, our ddPCR analyses on positive controls were clearly positive for both HPV16 and HPV18 DNA, and thus it is highly unlikely that genetic changes are the explanation for the ccfHPV-DNA-negative cases.

However, it is important to keep in mind that the assay used ought to be of clinical significance, meaning that it should only detect clinically relevant ccfHPV DNA. In one of the prior studies [35], ccfHPV DNA positivity was defined as ≥0 copy/mL plasma, and another study established cutoff by means of HPV positive tissue samples and defined ccfHPV DNA positivity as ≥1 droplet at the same amplitude as the positive control [31]. Our results on ccfHPV DNA in the 15 healthy controls showed that some of these patients had 1-2 HPV-positive copies per mL. A potential explanation for these positive droplets is the possible cross-reaction with human genes with similar sequences as the primer sequences used for ccfHPV DNA detection. The addition of specific probes assists in preventing these cross-reactions, but when running multiple PCR cycles, it is likely that a few positive copies may occur, but this does not mean that the sample is positive for ccfHPV DNA. Another particularly important factor in regard to performing HPV analyses may be low level contamination, either from other patient samples handled and analyzed simultaneously or from the person handling the samples. Studies having examined the contamination potential of HPV have established that HPV DNA is detectable in up to 18% of samples obtained from fomites in a gynecological outpatient clinic [54], and that HPV DNA particles deposited on environmental surfaces may stay infectious for up to seven days after desiccation [55,56]. It is therefore exceedingly important that all procedures regarding the handling of samples follow a strict and sterile protocol. Once again, it is therefore central to underline the importance of including negative control samples in the assay, both in order to establish a reliable cutoff for ccfHPV DNA positivity, but not least to ensure that contamination has not occurred.

Interestingly, none of the 12 patients with HPV18-related cervical cancer in this study were ccfHPV18 DNA positive (Figure 2 and Appendix A). This is in line with findings from Kang et al. [37], showing a large variation among cancer patients. Here, the median level of ccfHPV DNA was 21,600 copies/mL for HPV16 and only 1360 copies/mL for HPV18 (*p =* 0.040). This was also observed in the study by Rungkamoltip et al. [36]. Here, the detection for ccfHPV DNA was 20.51% for HPV16 and only 10.26% for HPV18. It should be noted though that the study by Rungkamoltip et al. [36] did not pre-select patients with HPV16- or HPV18-associated carcinomas. Their data would therefore more likely reflect the probability of using ccfHPV16 or ccfHPV18 DNA to identify cervical cancer patients from a general population. A notable difference between these studies and our study is that even though they find that both detection rate and ccfHPV DNA level is much lower for HPV18-associated patients, they are actually able to detect ccfHPV18 DNA in some patients. As described earlier, sensitivity of the assays and not least the chosen cutoff for ccfHPV DNA positivity may contribute to these differences. Thus, to exclude a possible detection problem, both false negatives and false positives, this study entailed various control components: (1) primers and probes were validated using DNA purified from cervical biopsies from selected cases with HPV16- or HPV18-associated disease (positive controls), (2) cfDNA from 15 healthy women were analyzed to determine a conservative and clinically relevant cutoff for ccfHPV DNA positivity, (3) all ddPCR analyses included an HPV-positive control sample (positive controls), and (4) all plasma samples were checked for the presence of human DNA. Furthermore, 9 of the 12 cases with HPV18-associated cervical cancer were early stage patients, which in both the present study and prior similar studies [31,35,36,37] correlate to either low levels of ccfHPV DNA or ccfHPV DNA negativity, regardless of HPV genotype.

Another possible explanation for the genotype specific variation in ccfHPV DNA findings may be differences in biological characteristics between HPV genotypes. As mentioned earlier, cervical cancer cells comprise an integrated proportion of HPV DNA containing the E6 and E7 oncogenes and/or entire viral episomes. Presence of viral DNA in the episomal state has been observed in approximately 40% of HPV16-positive tumors [26], whereas the integrated state has been observed in almost all HPV18-positive tumors [57,58]. Since prior research has shown that patients with episomal HPV have a higher viral load compared to patients with integrated HPV [59], one possible explanation for the much lower rate of ccfHPV18 DNA detection may be integration and, accordingly, lower viral load. Integration may therefore also be the cause for some of the ccfHPV16-negative patients. Overall, to support this hypothesis as well as all other findings and tendencies of the present study, a larger patient cohort is needed.

## 5. Conclusions

In conclusion, our study showed that ccfHPV DNA may not be a promising marker for early detection of cervical cancer, but for patients with advanced stage disease, a quantitative assessment of ccfHPV DNA reflects tumor burden and may therefore be useful for establishing treatment response and monitoring the disease.

## Figures and Tables

**Figure 1 cells-11-02170-f001:**
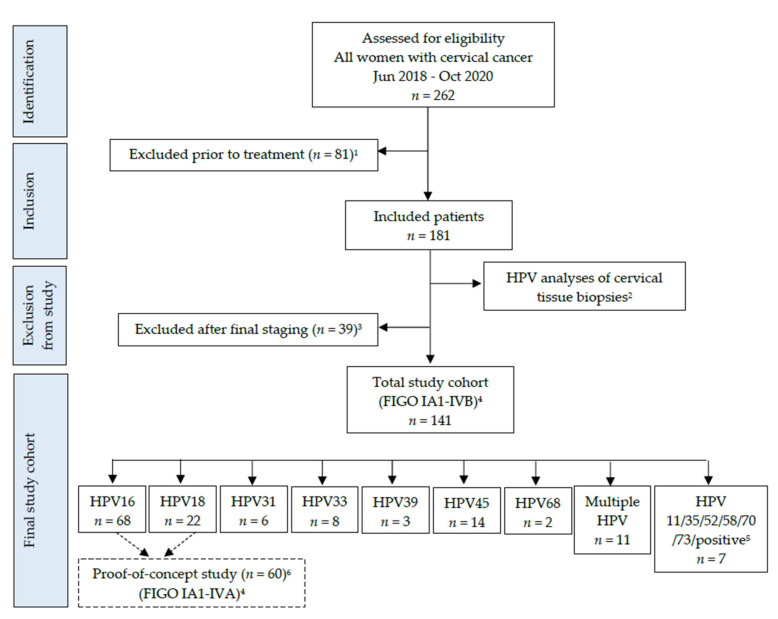
Flowchart of included cervical cancer patients. ^1^ For exclusion causes, see Appendix A. ^2^ Cervical biopsies were analyzed with the INNOLiPA HPV Genotyping Extra II assay (INNO-LiPA) (Fujirebio Europe, Ghent, Belgium). ^3^ For exclusion causes, see Appendix A. ^4^ International Federation of Obstetrics and Gynecology, FIGO 2018 (see ref [48]): All the mentioned HPV genotypes were only detected in samples from one patient each. One sample was categorized as “HPV positive” by INNOLiPA, which means that the sample is positive for one or more of a broad range of mucosal HPV genotypes other than the 32 genotypes specifically detected by the assay. The assay is therefore unable to genotype this one sample. ^6^ In the proof-of-concept study, the first 30 patients with HPV16/18 positive early stage cancer and the first 30 patients with HPV16/18 po-sitive advanced stage cancer were included.

**Figure 2 cells-11-02170-f002:**
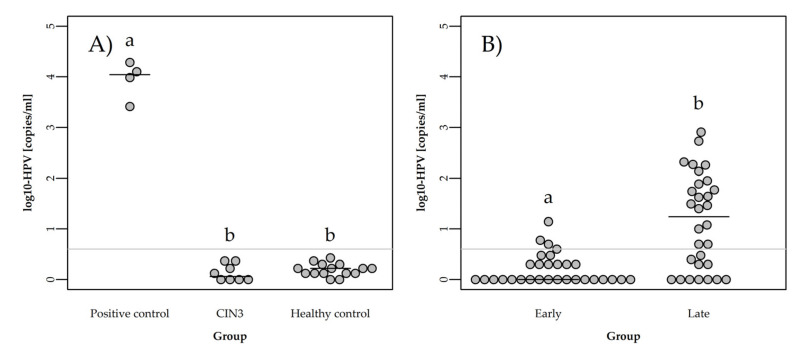
Detection of ccfHPV16/18 DNA by ddPCR in (**A**) four cervical cancer tissue samples (po-sitive control), eight plasma samples from CIN3 patients (CIN3), and 15 plasma samples from healthy individuals (negative control), and in (**B**) plasma samples from the early stage and the late-stage subgroup, respectively. Grey line represents cutoff value for HPV positivity (<3 copies/mL). Different letter (a,b) denote statistically different groups. Black lines represent median values.

**Figure 3 cells-11-02170-f003:**
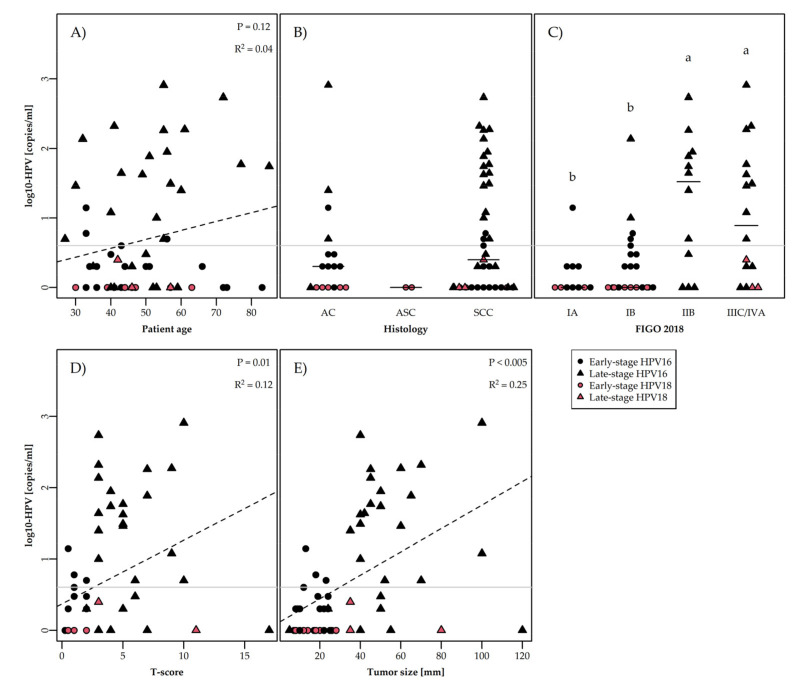
Concentration (copies/mL) of ccfHPV DNA (log scale) detected by ddPCR in cervical cancer patients according to clinical and biological criteria. Statistical significance was tested using the Kruskall–Wallis test followed by Conover–Iman for (**B**,**C**) and linear regression for (**A**,**D**,**E**). Grey line represents cutoff value for ccfHPV DNA positivity (>3 copies/mL). In (**C**), different letters (a,b) denote statistical difference between two groups, whereas the same letters (a, a or b, b) denote no statistical difference between two groups. In (**B**,**C**), black lines represent median values. (**A**) Concentration by patient age. (**B**) Concentration by histology (AC = adenocarcinomas, ASC = adenosquamous carcinoma, SCC = squamous cell carcinoma). (**C**): Concentration by FIGO 2018 stage (IA = IA1 and IA2, IB = IB1, IB2, and IB3, and IIIC/IVA = IIIC1, IIIC2, and IVA). (**D**) Concentration by tumor score (T score)^1^. (**E**) Concentration by tumor size (mm). T score was evaluated with the TS system presented in Table 1 in Lindegaard, J.C., et al. (see ref [46]).

**Table 1 cells-11-02170-t001:** Patient and tumor characteristics for the two subgroups (*N =* 60).

Patients	Early Stage Subgroup, *N* = 30	Late-Stage Subgroup, *N* = 30
Age (mean ± SD)	47 ± 13	51 ± 13
Histology ^1^	***n* (%)**	***n* (%)**
SCC	15 (50.0)	26 (86.7)
ASC	2 (6.7)	
AC	13 (43.3)	4 (13.3)
FIGO 2018 ^2^	***n* (%)**	***n* (%)**
IA1	1 (3.3)	
IA2	10 (33.3)	
IB1	7 (23.3)	
IB2	12 (40.0)	
IB3		2 (6.7)
IIB		12 (40.0)
IIIC1		13 (43.3)
IIIC2		2 (6.7)
IVA		1 (3.3)
T score (mean ± SD) ^3^	1.2 ± 0.7	5.6 ± 3.3
Tumor size (mm) (mean ± SD) ^4^	15.4 ± 6.6	49.83 ± 19.7
HPV genotype tissue ^5^	***n* (%)**	***n* (%)**
16	21 (70.0)	27 (90.0)
18	9 (30.0)	3 (10.0)

^1^ Abbreviations: squamous cell carcinoma (SCC), adenocarcinoma (AC), adenosquamous carcinoma (ASC). ^2^ Disease stage according to FIGO 2018 (see ref [48]). For early stage patients having been re-staged after surgery, the re-staged stage is the one listed. For primary stage, see Appendix A. ^3^ T score according to Lindegaard, J.C., et al. (see ref [46]). The scoring system is developed for cervical cancer patients with stage IB-IVB disease, giving patients with stage IB1, IB2, and IB3 T scores of 1, 2, and 3, respectively. Thus, for patients with stage IA1 and IA2 disease, we made a presumption that stage IA1 equals a T score of 0.25 and stage IA2 equals a T score of 0.5. For early stage patients having been re-staged after surgery, the re-staged stage is the one used to determine T score. For primary disease stage, see Appendix A. ^4^ Large diameter of tumor. For the early stage subgroup, tumor size was evaluated pathologically after surgery on the basis of the removed tissue. For the late-stage subgroup, tumor size was evaluated on the basis of clinical examination or magnetic resonance imaging (MRI) prior to treatment. ^5^ Cervical tissue biopsy tested with INNOLiPA^®^ Genotyping Extra II (Fujirebio).

## Data Availability

Data are contained within the article.

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
