# Peer review of "The Diagnostic Value of Circulating Cell-Free HPV DNA in Plasma from Cervical Cancer Patients"

_cells, 2022, doi:10.3390/cells11142170_

Round 1
Reviewer 1 Report
In this manuscript, Sara and colleagues have performed a retrospective analysis on plasma samples collected from cervical cancer patients with different stages of the disease, patients with CIN3, and healthy controls. In this study they quantified cell-free HPV DNA using a highly sensitive digital droplet PCR. They showed that cell-free HPV DNA can be a promising tool in advanced stage cervical cancer to establish tumor burden. It can be used for determining therapeutic response and disease progression. Overall, the manuscript is well written and the discussion section is quite comprehensive.
Author Response
Dear reviewer,
Thank you for your useful inputs and observations, they are greatly appreciated.
Reviewer 2 Report
This is an interesting and technically well-cared manuscript. As cited, there have been several publications on the subject, but the careful application of the methodology and accurate attribution of cutt-offs put this manuscript at a different quality level and general interest. My only comment goes to the lack of detection of ccfHPV18 DNA. I tend to agree with the argument of HPV DNA integration; however, the authors must discuss if additional primers and probes were tested and validated to exclude a detection problem. I beleive this is a relevant aspect particularly when considering risk and prognosis for adenocarcinomas as compared to squamous cell carcinomas.
Author Response
Dear reviewer,
Thank you for your useful inputs and observations, they are greatly appreciated. Regarding the ccfHPV18 DNA negative findings, your comment is very relevant and needs to be addressed. We believe that our ddPCR assay included several components that can exclude a detection problem. 1) Firstly, cervical tissue DNA from selected cases from the cohort was used to validate the primers and probes used for both HPV genotypes and also to exclude the possibility that genetic changes could explain the ccfHPV DNA negative cases. 2) Secondly, all ddPCR analyses included an HPV16 or HPV18 positive control consisting of the same cervical tissue DNA as decribed in 1). 3) Lastly, by chance, nine of the 12 patients with HPV18 associated cervical cancer were early-stage patients, and as is already well-discussed in the manuscript, both this study and prior studies find that the disease stage correlate to either low levels of ccfHPV DNA or ccfHPV DNA negativity, which may therefore be the simple explanation for these ccfHPV18 DNA negative findings.
To emphazise what we did to exclude a detection problem, the following additions to the discussion section will be suggested to the editor:
- Line 397-400: Each ddPCR run included a non-template control, cfDNA from a healthy control and an HPV positive control consisting of purified DNA from cervical biopsies from selected cases with HPV16 and cases with HPV18 associated cervical cancer (positive controls).
- Line 451-454:
In the present study, our ddPCR analyses on positive controls were clearly positive for both HPV16 and HPV18 DNA, and thus, it is highly unlikely that genetic changes are the explanation for the ccfHPV DNA negative cases.
- Line 490-500: Thus, to exclude a possible detection problem, both false negatives and false positives, this study entailed various control components; 1) primers and probes were validated using DNA purified from cervical biopsies from selected cases with HPV16 or HPV18 associated disease (positive controls), 2) cfDNA from 15 healthy women were analyzed to determine a conservative and clinically relevant cutoff for ccfHPV DNA positivity, 3) all ddPCR analyses included an HPV positive control sample (positive controls), and 4) all plasma samples were checked for presence of human DNA. Furthermore, nine of the 12 cases with HPV18 associated cervical cancer, were early-stage patients, which in both the present study and prior similar studies [32, 36-38] correlate to either low levels of ccfHPV DNA or ccfHPV DNA negativity regardless of HPV genotype.
This manuscript is a resubmission of an earlier submission. The following is a list of the peer review reports and author responses from that submission.